# Outcomes of a primary care mental health implementation program in rural Rwanda: A quasi-experimental implementation-effectiveness study

Stephanie L. Smith[1,2,3,4]*, Molly F. Franke[3], Christian Rusangwa[4], Hildegarde Mukasakindi[4], Beatha Nyirandagijimana[4], Robert Bienvenu[4], Eugenie Uwimana[5], Clemence Uwamaliya[5], Jean Sauveur Ndikubwimana[5], Sifa Dorcas[4], Tharcisse Mpunga[5], C. Nancy Misago[6], Jean Damascene Iyamuremye[6], Jeanne d'Arc Dusabeyezu[6], Achour A. Mohand[6], Sidney Atwood[3], Robyn A. Osrow[4], Rajen Aldis[4], Shinichi Daimyo[1], Alexandra Rose[1], Sarah Coleman[1], Anatole Manzi[1,4], Yvonne Kayiteshonga[6], Giuseppe J. Raviola[1,3]

1 Partners In Health, Boston, MA, United States of America, 2 Department of Psychiatry, Brigham and Women's Hospital, Boston, MA, United States of America, 3 Department of Global Health and Social Medicine, Harvard Medical School, Boston, MA, United States of America, 4 Partners In Health/Inshuti Mu Buzima, Kigali, Rwanda, 5 Ministry of Health, Kigali, Rwanda, 6 Mental Health Division, Rwanda Biomedical Center, Kigali, Rwanda

* stephanie_smith@hms.harvard.edu

**Data Availability Statement:** Data cannot be shared publicly because of permission for public dissemination was not approved by the Rwanda

## Abstract

### Introduction

To address the know-do gap in the integration of mental health care into primary care in resource-limited settings, a multi-faceted implementation program initially designed to integrate HIV/AIDS care into primary care was adapted for severe mental disorders and epilepsy in Burera District, Rwanda. The Mentoring and Enhanced Supervision at Health Centers (MESH MH) program supported primary care-delivered mental health service delivery scale-up from 6 to 19 government-run health centers over two years. This quasi-experimental study assessed implementation reach, fidelity, and clinical outcomes at health centers supported by MESH MH during the scale up period.

### Methods

MESH MH consisted of four strategies to ensure the delivery of the priority care packages at health centers: training; supervision and mentorship; audit and feedback; and systems-based quality improvement (QI). Implementation reach (service use) across the 19 health centers supported by MESH MH during the two year scale-up period was described using routine service data. Implementation fidelity was measured at four select health centers by comparing total clinical supervisory visits and checklists to target goals, and by tracking clinical observation checklist item completion rates over a nine month period.

A prospective before and after evaluation measured clinical outcomes in consecutive adults presenting to four select health centers over a nine month period. Primary outcome

National Ethics Committee. Data are available from the Rwanda National Ethics Committee (contact via info@rnecrwanda.org) for researchers who meet the criteria for access to confidential data.

**Funding:** This study was generously funded by the Abundance Foundation and Rick and Nancy Moskovitz. The funders had no role in study design, data collection and analysis, decision to publish, or preparation of the manuscript. Partners In Health, a non-profit 501(c)(3) organization, provided support in the form of salaries for authors [SLS, CR, HM, BN, RB, SD, RAO, RA, SD, AR, SC, AM, GJR], as well as financial support for health service delivery in Burera district, but did not have any additional role in the study design, data collection and analysis, decision to publish, or preparation of the manuscript. The specific roles of these and all authors are articulated in the 'author contributions' section.

**Competing interests:** The authors declare no competing interests. Authors SLS, CR, HM, BN, RB, SD, RAO, RA, SD, AR, SC, AM, and GJR are affiliated with and/or employed by the non-profit 501(c)(3) organization Partners In Health. This affiliation does not alter our adherence to PLoS ONE policies on sharing data and materials.

assessments at baseline, 2 and 6 months included symptoms and functioning, measured by the General Health Questionnaire (GHQ-12) and the World Health Organization Disability Assessment Scale (WHO-DAS Brief), respectively. Secondary outcome assessments included engagement in income generating work and caregiver burden using a quantitative scale adapted to context.

## Results

A total of 2239 mental health service users completed 15,744 visits during the scale up period.

MESH MH facilitated 70% and 76% of supervisory visit and clinical checklist utilization target goals, respectively. Checklist item completion rates significantly improved overall, and for three of five checklist item subgroups examined. 121 of 146 consecutive service users completed outcome measurements six months after entry into care. Scores improved significantly over six months on both the GHQ-12, with median score improving from 26 to 10 (mean within-person change 12.5 [95% CI: 10.9–14.0] p< 0.0001), and the WHO-DAS Brief, with median score improving from 26.5 to 7 (mean within-person change 16.9 [95% CI: 14.9–18.8] p< 0.0001). Over the same period, the percentage of surveyed service users reporting an inability to work decreased significantly (51% to 6% (p < 0.001)), and the proportion of households reporting that a caregiver had left income-generating work decreased significantly (41% to 4% (p < 0.001)).

## Conclusion

MESH MH was associated with high service use, improvements in mental health care delivery by primary care nurses, and significant improvements in clinical symptoms and functional disability of service users receiving care at health centers supported by the program. Multifaceted implementation programs such as MESH MH can reduce the evidence to practice gap for mental health care delivery by nonspecialists in resource-limited settings. The primary limitation of this study is the lack of a control condition, consistent with the implementation science approach of the study.

## Study registration

ISRCTN #37231.

## Introduction

Mental disorders account for a substantial portion of the burden of illness across the globe. [1] In low- and middle-income countries (LMICs) with few specialized mental health providers, task sharing—moving care responsibilities from more specialized providers to less specialized health workers—has been increasingly emphasized as a key approach for addressing the global burden of mental disorders. [2–5] Evidence for the clinical effectiveness of task sharing in mental health care for a range of mental disorders is now clear. [6–9] From this evidence, the World Health Organization (WHO) has developed guidelines and cost-effective care packages for non-specialist providers to manage priority mental disorders in a variety of settings. [10]

Although task sharing packages for mental health care have been articulated, little evidence exists on how to turn these care packages into sustained service delivery systems in resource-limited and complex health care settings, particularly at scale. Implementation of standardized mental health care packages in any setting requires multi-faceted strategies that are guided by local experience and actively engage providers. [11] An effective implementation program for task sharing in mental health care must address both clinician- and systems-level factors, as most non-specialized health workers have had little or no experience in basic mental health assessment and treatment planning, or mental health service delivery [12–13]. General health system structures are often not designed to incorporate mental health care and supervision into settings with multiple competing clinical priorities and few human and financial resources. Implementation programs which focus on the general clinical acumen of front-line providers and which establish accountability for the quality of mental health care provision, as well as augment general health systems' capacity to incorporate mental health service delivery, are critical for effective task shared mental health care package implementation and scale-up in resource-limited settings.

With a population of 11.2 million people, the Republic of Rwanda is 159[th] on the human development index despite very significant growth in various public and private sectors over the past two decades. Rwanda has also had to contend with significant mental health burdens, including high rates of depression and posttraumatic stress related disorders (PTSD) due to the 1994 genocide, limited access to mental health services, and high levels of untreated severe mental illness. [14] In recognition of this high burden, over the past two decades, mental health services have been gradually decentralized, with mental health services increasingly provided by at least one psychiatric nurse and psychologist in each of 42 district hospitals. [15] However, the national budget for mental healthcare, as well as the ratio of public mental health workers to population within each district, remains very low. While decentralization of mental health services—the dissemination of nonspecialist-delivery care at district general hospitals, primary care health centers and in communities in an organized way—is a core tenet of the national mental health plan, the capacity of the government to implement decentralization has been limited due to significant resource constraints.

One potential solution for the challenges of mental health service decentralization and the scale-up of non-specialist delivered mental health care packages is to use effective implementation strategies designed for other clinical endeavors as a starting point. [15] Partners In Health (PIH), a nonprofit organization working in ten countries, has been working to support the public health delivery system in three rural districts of Rwanda since 2005, in close collaboration with its local sister organization, Inshuti Mu Buzima (IMB). In 2010, the Rwandan MoH and PIH collaboratively launched the Mentoring and Enhanced Supervision at Health Centers (MESH) program, an implementation program designed to strengthen the primary care system and improve the quality of care provided by nurses at primary care health facilities in PIH-supported districts of Rwanda. [16] The program's initial focus was to support health center nurse implementation of clinical protocols in child health, obstetrical and neonatal care, HIV care, and adult health. Results from health centers supported by the MESH program demonstrate significant improvements in a number of quality of care indicators following implementation [16–17].

In 2012, PIH/IMB, in collaboration with the Rwandan MoH, adapted MESH for mental health care delivery. MESH for Mental Health (MESH MH), is a multifaceted implementation program using four primary implementation strategies: decentralized training of primary care nurses in evidence-based mental health care packages; clinical mentoring of primary care nurses at rural satellite health centers by an experienced government psychiatric nurse; continuous primary care nurse clinical performance audit and feedback; and collaborative facilitation

of systems-based Quality Improvement (QI) projects. MESH MH was designed to operate within the existing MoH infrastructure, and clinical, supervisory and administrative staff for the program are located within the public sector. PIH/IMB together with the Mental Health Division of the MoH and funded by Grand Challenges Canada, scaled primary care delivered mental health services from six to nineteen health centers across Burera district between 2014 and 2016, using the MESH MH implementation program to facilitate this scale-up.

Implementation science studies are needed to address the significant knowledge gap between evidence based care packages for mental disorders and their actual delivery, especially in resource-limited settings. To this end, we conducted a quasi-experimental study to assess the implementation reach, fidelity, and clinical effectiveness of our primary care delivered mental health services, supported by MESH MH, as the services were scaled within one rural district of Rwanda.

## Methods

This evaluation was approved by the Rwanda National Ethics Committee (Protocol #736/ RNEC/2016) and deemed exempt by the Harvard University Institutional Review Board.

The initial protocol for this study is published at doi: 10.1136/bmjopen-2016-014067 [18] and is available in S1 and S2 Files.

### Study setting

The MESH MH implementation program was adapted in 2012 from the larger MESH program, and used initially at six health centers over two years in Burera district, which has a population of approximately 340,000. Between 2014 and 2016, mental health services at primary care centers were rolled out sequentially from 6 to 19 health centers supported by the MESH MH implementation program, for full district coverage. In addition to the 19 primary care health centers, the district health system also includes a 150-bed general hospital (Butaro Hospital), which operates as the district referral center for acute medical and psychiatric problems, and houses an outpatient speciality mental health clinic staffed by several government psychiatric nurses, and a government psychologist. The district health system also includes approximately 1500 Community Health Workers (CHWs) who are based within villages and affiliated with health centers as community linkages to the formal health system.

### Evidence-based care packages

Four major mental and neurologic disorders were chosen for initial clinical focus based on clinical priorities and disease burden perceived by district mental health staff and health center directors within their catchment areas during the early stages of MESH MH development: schizophrenia; bipolar disorder; major depressive disorder; and epilepsy. Care packages for each disorder were designed for delivery at rural primary care health centers based on feasibility and evidence of treatment effectiveness [10], in collaboration with mental health providers within the district (Table 1).

### MESH MH implementation program

The MESH MH Implementation Program consisted of four basic strategies to ensure the delivery of the priority care packages at health centers: Training, Supervision and Mentorship, Audit and Feedback, and Systems-Based Quality Improvement (QI) (Table 2).

**Strategy 1: Training.** 38 primary care nurses participating in the MESH MH program received an initial forty-hour training focused on the care packages, taught by government

**Table 1. Components of health center delivered mental health care packages.**

| |
|---|
| Complete mental health assessment, including medical and psychosocial assessment |
| Psychoeducation to service users and families |
| Psychosocial interventions: |
| - Address psychosocial stressors (all disorders) |
| - Behavioral activation (depression) |
| - Sleep hygiene and coping strategies (bipolar disorder/psychotic disorders) |
| - Facilitated rehabilitation in collaboration with CHWs (psychotic disorders) |
| Medication management |
| Regular monitoring and follow up (weekly to monthly) |
| Referral to community-based support for adherence promotion and follow-up management as needed. |
| Triage and referral to specialist mental health care for acute or complex needs as needed |

psychiatric nurses based at Butaro hospital. The training curriculum focused on performing an accurate general diagnostic assessment of persons presenting with mental health concerns, including medical evaluation of presenting symptoms, crisis interventions, and referral pathways for the selected major mental disorders and epilepsy. Nurses were also taught to generate and carry out a plan of care appropriate for each service user, including pharmacology as needed, psychoeducation, and linkages with social and community-based supports. The training also included sessions designed to improve nurses' general clinical acumen such as communication skills and developing rapport with service users, managing challenging emotional situations, and responding effectively to a variety of challenging clinical situations. The training curriculum was based on existing MoH and PIH guidelines as well as the World Health Organization Mental Health Gap Action Program (mhGAP). [10] Brief refresher trainings focused on areas of perceived need by government psychiatric nurse-mentors, were held every six months.

MESH MH also included basic training for Community Health Workers (CHWs) in case finding, treatment adherence, psychoeducation and stigma reduction. Training for one CHW in each village (approximately 750 total CHWs) began several months after MESH MH mental health supported services were incorporated at health centers. CHWs were supported by a PIH community coordinator and a public community health nurse at each participating health center.

**Table 2. MESH MH implementation strategies.**

| Strategy | Description |
|---|---|
| 1. Training | - Decentralized, interactive training for primary care nurses and community health workers on priority care packages, including the distribution of educational materials |
| | - Refresher trainings held every six months |
| 2. Clinical Supervision and Mentorship | - Primary care nurses provided with ongoing supervision and mentorship focused on care package implementation. |
| | - Psychiatric nurse-mentors provided with training on best practices in supervision. |
| | - Established program goal included three supervisory visits to each health center per month for at least six months, then bimonthly visits for six months, and monthly visits after one year of supervision |
| 3. Audit and Feedback | - Clinical performance data from a structured checklist was collected from at least three observed interviews per supervisory visit |
| | - Performance data was shared with primary care nurses to monitor, evaluate, and modify clinical practice |
| 4. Systems-Based Quality Improvement | - System based changes at health centers were implemented in a cyclical fashion using small tests of change (PDSA cycles). |

**Strategy 2: Supervision and mentorship.** Immediately following the training, health centers newly participating in the MESH MH program established a weekly mental health clinic day. On that day, each participating primary care nurse received a clinical supervisory-mentorship visit by a psychiatric nurse-mentor from the district hospital. The program goal was to complete three supervisory visits to each health center per month for at least six months, then bimonthly visits for six months, and monthly visits after one year of supervision. Supervisory visit targets were determined during the initial piloting of MESH MH (2012–2014) prior to the district scale up. Supervisory visits included direct clinical observation, individual case review, documentation review, and brief didactic sessions on relevant topics. MESH MH psychiatric nurse-mentors also participated in organizational-wide trainings for designed to improve supervisory and mentorship skills among mentors across clinical domains.

**Strategy 3: Audit and feedback.** A clinical checklist was developed to track each nurse's provision of mental health care at health centers. The goal was to complete a checklist for at least three clinical encounters during each supervisory session by a MESH MH mentor. The checklist contained dichotomous scoring of key observable features of basic mental health evaluations, including aspects of assessment, treatment and follow-up planning, and referral procedures. The nurse mentor used the checklist to ensure that health center nurses were performing basic mental health evaluations, accurately diagnosing service users, and offering appropriate treatment including both psychopharmacologic management of symptoms as needed, as well as an appropriate choice of psychoeducational and behavioral interventions for the clinical situation. Mentors and nurses discussed checklist scores on a regular basis, and primary care nurses were provided specific feedback on clinical strengths and areas for improvement based on their checklist scores.

**Strategy 4: Systems-based QI.** During each supervision session, the nurse-mentor also used a structured QI process, a short-term rapid learning approach, to facilitate system improvements for primary care-delivered mental health care. The MESH QI process used continuous plan-do-study-act (PDSA) cycles, which devised health center derived indicators, identified specific addressable problems, and implemented, monitored, and modified solutions as needed based on the chosen indicators. [19] The MESH MH nurse-mentor used the QI process to stimulate discussion with a designated clinical and administrative team at each health center around systems-based performance issues and mental health care "quality gaps". After gaps were identified, the mentor worked together with the health center staff to formulate and record the specific solutions to improving quality gaps, in order to return to them frequently until resolutions were found.

## Data collection: Implementation reach (service use)

Routine program monitoring data were collected from paper registries for all service users attending mental health services at all health centers during the two-year scale up period, from 6 to 19 health centers supported by MESH MH. Visit data are recorded in the daily register by each clinician at all health facilities in the district. Data officers collected routine service user variables from the paper registers and entered them into a centralized database (Microsoft Corp, Redmond, Washington, USA). Collected variables included the total number of unique service users seen for a mental disorder or epilepsy, and the total number of recorded visits for a mental disorder or epilepsy.

## Data collection: Implementation fidelity

All MESH MH mentor clinical observation checklists completed during each supervisory and quality improvement visit were collected from the MESH MH nurse mentor, reviewed for

accuracy and completeness by a research officer, and entered into a database. Four health centers were chosen for convenience to measure implementation fidelity as those health centers were newly participating in the scale-up of health services supported by MESH MH. Implementation fidelity was measured by 1) comparing the total number of supervisory and quality improvement visits completed each month from November 2014 to July 2015 relative to the target number (96 total visits during the nine month study period, to meet the program goal of three monthly visits to each health center for the first six months, then two monthly visits for the subsequent three months) and 2) comparing the number of checklists completed during each supervisory visit relative to the target number (288, to meet the program goal of at least three directly observed clinical interviews per supervisory visit during the nine month period). Implementation fidelity was also measured over the same time period by tracking the mean proportion of successfully completed checklist items each month (overall and on each of five subsets of clinical observation checklist items, Table 3).

### Data collection: Clinical effectiveness

**Participants.** Between November 1st 2014 and July 1st 2015, consecutive adults (age 18 and over) diagnosed with any mental disorder or epilepsy at four health centers newly participating in the MESH MH program were enrolled in a prospective, implementation science-driven evaluation of primary care nurse delivered mental health services. MESH MH program scale-up occurred in cycles of four to five health centers every six months throughout the district scale-up period (2014 to 2016) until district coverage was complete. The four health centers receiving MESH MH service support at the beginning of the rollout's funding cycle (FY 2015) were chosen for logistical and timing convenience to measure clinical outcomes and implementation fidelity. The goal was to evaluate the clinical outcomes of service users receiving basic mental health care packages of care within the primary care settings supported by MESH MH. Written informed consent from each service user and his/her designated proxy was collected at enrollment. Persons who needed to be transferred to the district hospital for an acute medical or psychiatric emergency, people who presented with a primary alcohol or substance use disorder, or those who were not able to have a family member accompany them at initial enrollment were excluded as stipulated by the Rwandan National Ethics Committee.

**Clinical outcomes and economic outcome measures.** Service user symptom and functioning assessments were performed at baseline, two and six months by a trained clinician-researcher. Service users were assessed only on their return for a routine follow-up visit at the health center. For those who did not return for follow-up initially, a CHW in their village was contacted to call or visit the service user and encourage the person to return to care. Those who then returned for routine follow-up were re-interviewed at two and six months after their initial visit. All others were considered lost to follow-up.

Primary outcomes evaluated included clinical symptoms and functional disability. The General Health Questionnaire (GHQ-12) is a general measure of clinical symptoms of psychological distress frequently used in primary care settings. [20] This scale was chosen rather than a disorder-specific symptom scale given the anticipated diagnostic heterogeneity of the study population. The World Health Organization Disability Assessment Scale Brief (WHO-DAS II Brief) scale was chosen as a general measure of functioning and disability across a variety of domains relevant to mental and neurological illness. [21] Although neither scale had yet been validated specifically in Rwanda, both scales have demonstrated high levels of validity and reliability across multiple cultures and languages. [22–25] All instruments were translated into Kinyarwanda, back translated prior to implementation, and pilot-tested among a convenience sample of ten service users at Burera district (Butaro Hospital) outpatient mental health clinic,

**Table 3. Subset of checklist indicators.**

| Clinical Focus | Scored Checklist Items (completed yes/no) |
|---|---|
| Intake | 1. Did the nurse ask for patient contact information (full address and family name)? |
| | 2. Did the nurse ask the patient why he/she is at the health center? |
| | 3. Did the nurse ask how long and how frequently the presenting symptoms have been happening? |
| | 4. Did the nurse find out how the presenting symptoms are affecting the patient's ability to work, go to school, or other social functioning? |
| | 5. Did the nurse ask about current and past medical illness (in order to check yes they need to have assessed both past and current)? |
| | 6. Did the nurse take a complete psychiatric history? |
| | 7. Did the nurse ask about substance use/abuse? |
| | 8. Did the nurse take a family history? |
| Treatment planning: non-medication based | 1. Assess if patient/family is aware of the diagnosis, and if he/she is not, did they disclose? |
| | 2. Discuss at least two relevant psychoeducation facts with the patient (from training materials)? |
| | 3. Discuss at least two relevant psychoeducation facts with the family? |
| | 4. If treating for depression, did the nurse discuss behavioral activation? |
| | 5. If treating for bipolar disorder, did the nurse discuss sleep hygiene? |
| Treatment planning: medication management | 1. Based on symptoms, diagnosis, and any history of side effects, did the nurse prescribe the correct medication(s)? |
| | 2. Prescribe the correct dose of the medication(s) (see training book)? |
| | 3. Tell the patient how the medication will help? |
| | 4. Tell the patient how to take the medication? |
| | 5. Tell the patient about potential side effects? |
| Follow up treatment planning: non-medication based | All of above items, plus: |
| | 1. Assess current status of target symptoms of the diagnosed disorder? |
| | 2. Assess for development of any new symptoms? |
| | 3. Ask/assess current level of functioning? |
| | 4. Address all current symptoms and current level of functioning? |
| | 5. Provide psychoeducation (ref. to training materials)? |
| Follow up treatment planning: medication based | 1. Assess medication response? |
| | 2. Ask about side effects? |
| | 3. Address side effects appropriately? |
| | 4. Based on symptoms, diagnosis, and any history of side effects, did the nurse prescribe the correct medication(s)? |
| | 5. Prescribe the correct dosage of medication(s)? (based on training materials)? |

to ensure face validity. The Cronbach alpha demonstrated good internal consistency for both the GHQ-12 ($\alpha$ = 0.93) and the WHO-DAS Brief ($\alpha$ = 0.90).

Secondary outcomes included a scale of economic burden and service use, adapted from a scale used in other resource limited settings. [26] The economic questionnaire included items on self-perception of illness severity, medication adherence, household features and goods as listed on the Rwanda National Survey, national economic classification, level of engagement in income generating work, estimated number of hours of care support provided for the service user by family and informal caregivers, the number of lost days of work for service users as well as caregivers, and health care and associated costs incurred by the service user and family.

**Sample size.** The enrollment goal was at least 116 patients, the minimum sample size needed to estimate the percent of patients with a clinically significant (25%) decrease in GHQ-12 score with a 95% confidence interval of +/- 10%. Based on past follow up rates for studies in other clinical domains, it was estimated that 80% of patients would be retained in the study and analysed, and that 50% of those would experience a clinically significant improvement. [18]

**Statistical analysis.** For outcome measures, we calculated change in GHQ-12 and WHO-DAS II Brief scores after two and six months, relative to baseline, and tested whether this change was different from zero using a paired t-test. Because not all patients had follow-up assessments after two and six months, we conducted sensitivity analyses in which we carried forward the last available score for participants missing the given follow-up assessment. To examine changes in binary outcomes (e.g., income generating activities and support needed in daily activities) we conducted hypothesis testing using McNemar's test for paired data and repeated sensitivity analyses in which carried forward the last value for patients lacking a follow-up assessment of these outcomes. We examined factors associated with missing a follow-up assessment at six-months using chi-squared, Fisher's exact, and Wilcoxon rank-sum tests. To test whether checklist completion improved over time, we used mixed effects regression analyses, which included a random intercept for each nurse in order to adjust variances for multiple assessments per nurse.

## Results

### MESH MH implementation reach (service use)

2239 unique service users were seen at all health centers supported by the MESH MH program during the district scale up period from November 2014 to October 2016. A total of 15,744 visits for a mental disorder or epilepsy occurred during this same period.

### Fidelity of MESH MH implementation

Between November 2014 and July 2015, a total of 67 supervision and mentorship visits, and quality improvement meetings, were completed at the four health centers participating in the evaluation. This number represents 70% of the program target goal of 96 visits during the study period. Mentors filled out a total of 220 observed interview checklists during the same period, which represents 76% of the program target goal of 288 completed checklists.

Overall, the mean proportion of checklist items successfully completed increased significantly over time (p<0.0001), from 44% to 80% at the end of nine-months (Fig 1). Within sub-groupings of checklist items, we also observed improvements over time, and these improvements were statistically significant for intake scores (35% to 75%, p = 0.002), non-medication based follow-up scores (53% to 82%, p<0.0001), and medication based follow-up scores (48% to 80%, p<0.0001).

### Evaluation of participants and follow-up

Between November 2014 and July 2015, 146 people were enrolled in the MESH MH program evaluation. 4 people were excluded due to a need for referral to specialist services for acute psychiatric or medical needs. A flow chart detailing enrollment, follow-up and analysis is detailed in Fig 2. Participant characteristics are detailed in Table 4.

A total of 121 (83%) remained in care after six months and completed the program evaluation questionnaires. Approximately 60 people (50% of those remaining in care) were directly supported by CHWs in their return to care, either by verbal encouragement or direct

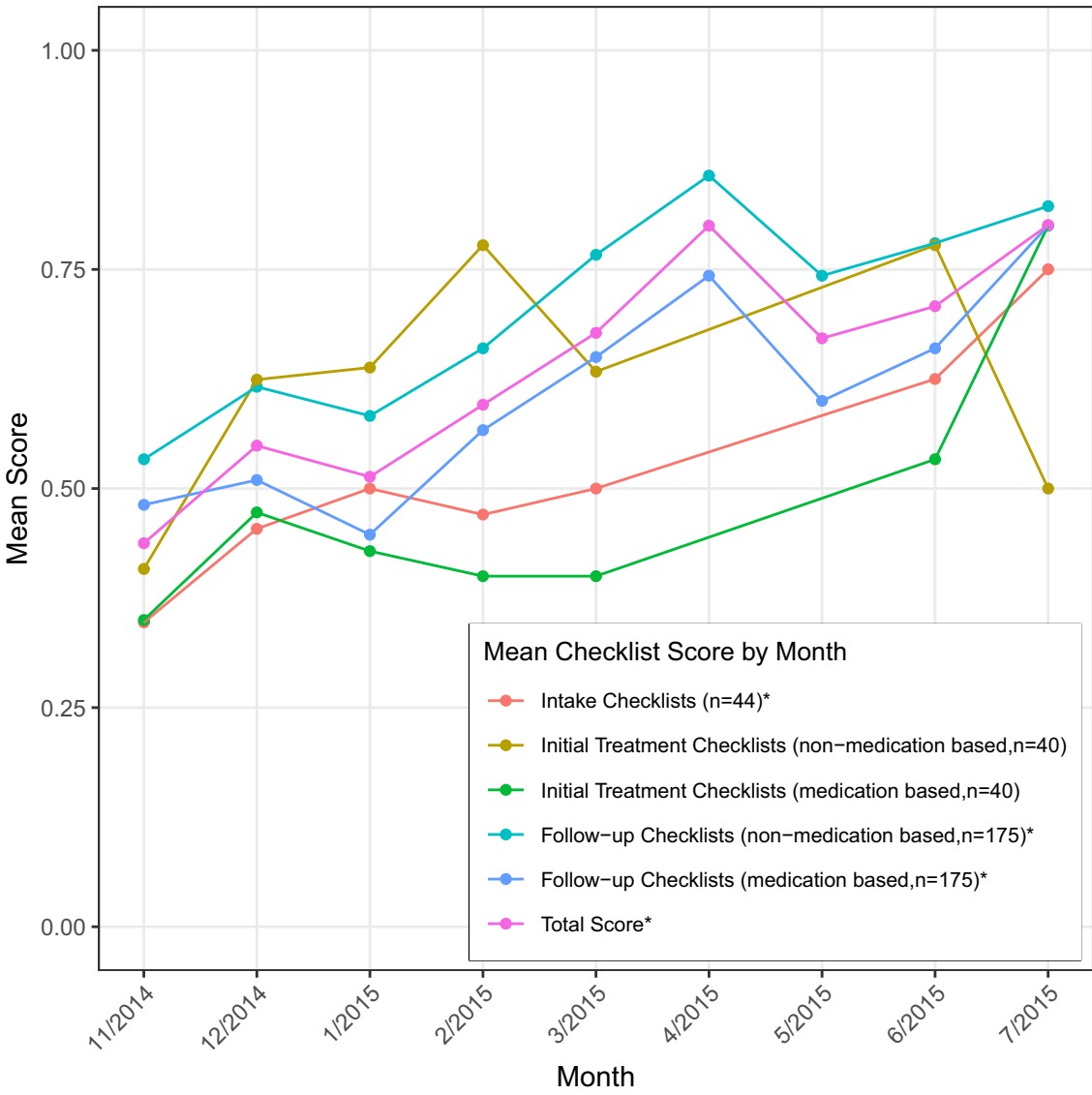

**Fig 1. Mean checklist score by month.** Dots indicate months in which checklist items were completed. Categories in the legend marked with an asterisk (*) showed significant improvement over nine months, all p<0.001 except Intake Score for which p = 0.002.

accompaniment to services. Older age, health center, owning no household items (cell phone, radio, bicycle or television), and a higher baseline GHQ-12 score were significantly positively associated with loss to follow-up (Table 5). For those who completed the evaluation, participants attended an average of 7.3 visits during the six-month study period.

## Clinical effectiveness: Symptoms and functioning

Over the six-month evaluation period, there was a significant improvement in score on the GHQ-12, with median score improving from 26 to 10 (mean within-person change 12.5 [95% CI: 10.9–14.0] p< 0.0001), and a significant improvement on the WHO-DAS Brief, with median score improving from 26.5 to 7 (mean within-person change 16.9 [95% CI: 14.9–18.8] p< 0.0001) (Table 6). Results were similar in sensitivity analyses in which we carried forward the last available score for patients who were missing the follow-up assessment.

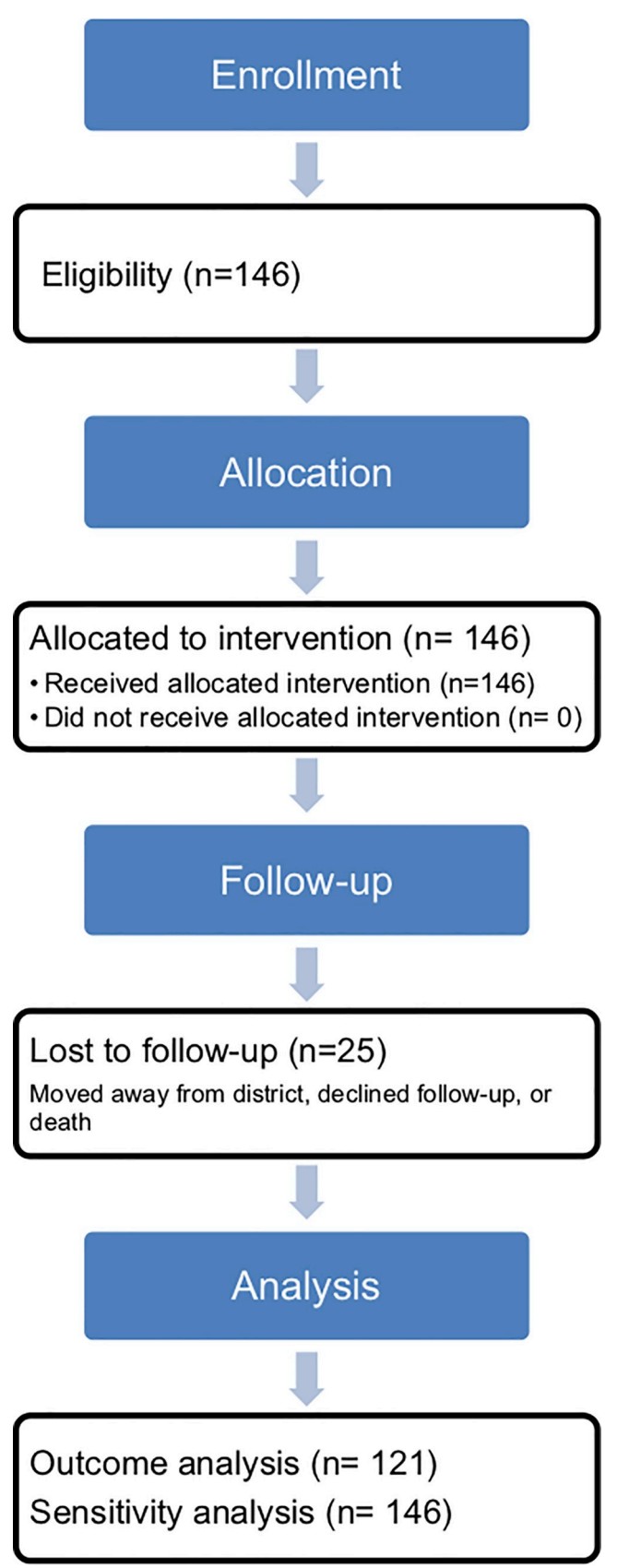

**Fig 2. Enrollment flow diagram, MESH MH evaluation.**

## Economic outcomes

There were significant decreases in the percentage of service users who reported an inability to work any day of the past thirty days (51% of surveyed service users at baseline to 6% after six months, p < 0.001), and the percentage of service users who reported the need for help with

**Table 4. Baseline characteristics of the service user cohort.**

|  | N (146) | % of total |
|---|---|---|
| **Female gender** | 96 | 66% |
| **Age** | | |
| 18–35 | 73 | 50% |
| 36–59 | 52 | 36% |
| 60 and up | 21 | 14% |
| **Education** | | |
| None | 57 | 39% |
| 1–3 years | 36 | 25% |
| 4–6 years | 41 | 28% |
| > 6 years | 12 | 8% |
| **Marital status** | | |
| Never married | 45 | 31% |
| Married | 68 | 47% |
| Separated | 16 | 11% |
| Widowed | 17 | 12% |
| **Health center** | | |
| A | 30 | 21% |
| B | 31 | 21% |
| C | 48 | 33% |
| D | 37 | 25% |
| **Employment** | | |
| Subsistence farming/labor | 86 | 59% |
| Non-income generating work | 15 | 10% |
| Labor | 3 | 2% |
| Studying | 6 | 4% |
| No productive work | 36 | 25% |
| **Has electricity** | 3 | 2% |
| **Water** | | |
| Protected Spring | 1 | 1% |
| Public tap | 84 | 58% |
| Unprotected spring | 40 | 27% |
| Surface water | 21 | 14% |
| **Sanitation** | | |
| Pit Latrine (non-shared) | 92 | 63% |
| Open pit | 50 | 34% |
| No facility | 4 | 3% |
| **Household goods** | | |
| Radio | 57 | 39% |
| Cellphone or telephone | 27 | 18% |
| Means of transport (bicycle, moto, car) | 8 | 5% |
| Television | 0 | 0% |
| None of the above | 79 | 54% |

**Table 5. Univariable predictors of missing follow-up evaluation at 6-months.**

| | Has follow-up | Missing follow-up | p-value |
|---|---|---|---|
| | (N = 121) | (N = 25) | |
| Female gender[a] | 82 (68) | 14 (56) | 0.26 |
| Age[a] | | | 0.02 |
| 18–35 | 64 (53) | 9 (36) | |
| 36–59 | 44 (36) | 8 (32) | |
| 60 or older | 13 (11) | 8 (32) | |
| Married[a] | 60 (50) | 8 (32) | 0.11 |
| Primary diagnosis[b] | | | |
| Bipolar disorder/Depression | 15 (12) | 2 (8) | 0.07 |
| Epilepsy | 22 (18) | 2 (8) | |
| Psychosis/Schizophrenia | 56 (46) | 19 (76) | |
| Other | 28 (23) | 2 (8) | |
| Health center[b] | | | 0.002 |
| A | 25 (21) | 5 (20) | |
| B | 26 (21) | 5 (20) | |
| C | 46 (38) | 2 (8) | |
| D | 24 (20) | 13 (52) | |
| No formal education[a] | 44 (36) | 13 (52) | 0.14 |
| No productive work[a] | 26 (21) | 10 (40) | 0.05 |
| Does not own radio, phone, television or mode of transport[a] | 60 (50) | 19 (76) | 0.02 |
| Median baseline GHQ-12 score[c] | 25 (17–32) | 30 (24–34) | 0.03 |

a. Chi-squared test

b. Fisher's exact test

c. Wilcoxon rank-sum test

activities of daily living (40% at baseline to 6% after six months, p < 0.001). There was also a significant reduction in the proportion of households who reported that a family member in the home had left income-generating work to care for the service user, from 41% at baseline, for a median of five days, to 4% after six months, for a median of two days (p < 0.001) (Table 7). Results were similar in sensitivity analyses in which we carried forward the last available value for patients who were missing the follow-up assessment.

## Discussion

Our study is among the first to describe implementation fidelity and outcomes of a multifaceted implementation program designed to capacitate government primary care health providers in a resource-limited setting to care for service users with severe mental disorders and epilepsy. The findings provide strong evidence that our cohort achieved significant clinical improvement relatively quickly, which was sustained through six months of treatment, including reductions in psychological distress and functional impairment. There were also significant household benefits to treatment, with decreases in caregiver burden, and increases in individual and family engagement in income-generating activities, during the study period. Our implementation fidelity outcomes show that adherence to the core components of the MESH MH strategies was high. These findings suggest that the MESH MH was generally delivered as intended, and nonspecialist primary health care workers at health centers successfully

**Table 6. Changes in GHQ-12 and WHO DAS Brief scores from baseline among service users.**

| | Baseline median score [IQR} | 2 months median score [IQR] | 6 months median score [IQR] | Mean within-person change (baseline to 2 months) | p-value for 2 month change | Mean within-person change (baseline to 6 months) | p-value for 6 month change |
|---|---|---|---|---|---|---|---|
| **GHQ-12 score** | | | | | | | |
| All available observations | 26 [18–33] | 12 [7–18] | 10 [6–17] | 11.6 [10.2, 13.9] | <0.0001 | 12.5 [10.9, 14.0] | <0.0001 |
| Last value carried forward | 26 18–33] | 14 [7–19] | 12 [6–18] | 10.1 [8.7, 11.4] | <0.0001 | 11.2 [9.7, 12.7] | <0.0001 |
| **WHO-DAS Brief score** | | | | | | | |
| All available observations | 26.5 [18–34] | 10 [1–18] | 7 [1–13] | 13.9 [12.0, 15.7] | <0.0001 | 16.9 [14.9, 18.8] | <0.0001 |
| Last value carried forward | 26.5 [18–34] | 12 [3–22] | 8 [2–16] | 12.1 [10.3, 13.8] | <0.0001 | 14.8 [12.9, 16.7] | <0.0001 |

NOTE: All available observations analyses include 146, 127, and 121 observations at baseline, 2 and 6 months, respectively. Last value carried forward analyses include 146 observations at each time point.

provided mental health care to service users with severe mental illnesses and epilepsy, leading to significant improvements in multiple domains.

We found that GHQ-12 scores and WHO-DAS Brief scores were relatively high at baseline, indicating high initial distress and disability in our participating population. Available data suggest that the highest proportion of disability associated with mental disorders occurs in LMICs. [27] Many people with severe mental disorders in those settings either receive no care, or receive care in central neuropsychiatric facilities disconnected from their communities. The clear and rapid improvements in distress and disability experienced by service users in our

**Table 7. Changes in income generation and daily activities among service users, and their caregivers.**

| | Baseline | 2 months | 6 months | p-value |
|---|---|---|---|---|
| | n (%) | n (%) | n (%) | (2-months vs. baseline) |
| **Inability to work <u>any day</u> in past 30 days as a result of a mental health condition** | | | | |
| All available observations[a] | 73 (51) | 15 (12) | 7 (6) | <0.0001 |
| Last value carried forward[b] | 73 (51) | 26 (18) | 17 (12) | <0.0001 |
| Requires help with activities of daily living | | | | |
| All available observations[c] | 59 (40) | 20 (16) | 7 (6) | <0.0001 |
| Last value carried forward[b] | 59 (40) | 28 (19) | 15 (10) | <0.0001 |
| **Primary caregiver left income- generating work to care for service user** | | | | |
| All available observations[d,e] | 60 (41) | 14 (11) | 5 (4) | <0.0001 |
| | (median: 5 days) | (median: 3.5 days) | (median: 2 days) | |
| Last value carried forward[b,f] | 60 (41) | 23 (16) | 14 (10) | <0.0001 |
| | (median: 5 days) | (median: 4 days) | (median: 4 days) | |

a. Analyses using all available observations include 144 baseline assessments, 125 2-month assessments, and 119 6-month assessments

b. Analyses using the last value carried forward include 146 observations at each time point

c. Analyses using all available observations include 146 baseline assessments, 124 2-month assessments, and 120 6-month assessments

d. Analyses using all available observations include 146 baseline assessments, 125 2-month assessments, and 120 6-month assessments

e. Observations for number of days of work missed were available for 59/60 people at baseline.

f. Observations for number of days of work missed were available for 59/60 people at baseline. Therefore, the last value carried forward analyses include 22 observations for this variable at 2 months and 13 at 6 months.

study demonstrates the urgency to capacitate rural general health systems in resource limited settings to care for individuals with severe mental illness close to their homes. Improving the capacity of local health care providers to successfully manage people with severe mental disorders may also increase family and community confidence in the local health system, thereby reducing reliance on central neuropsychiatric facilities.

At the end of our study period, we found significantly increased engagement in income generating activities by service users and a decreased reliance on household caregivers for activities of basic living. Engaging in productive work not only generates earnings, but also improves social participation, both of which are conducive to better mental health. [28] We also found increased engagement in income-generating activities by caregivers. Although the interplay between severe mental and neurologic disorders and poverty is complex, our data correlating mental health care delivery and improvements in the economic productivity of service users and caregivers are aligned with previous data documenting the positive effects of mental health interventions on the economic status of service users and families. [29] Care for severe mental disorders may play multiple roles in poverty reduction efforts in resource-limited settings, for example through fostering social capital for service users and families as well as increasing the ability of caregivers to be productive economically for the entire household.

Our MESH MH implementation model used four specific evidence-based strategies to help primary care nurses and health centers to implement care packages. Fidelity to the model was 70 and 76% of target goal for the total number of supervisory and QI visits, and completed clinical checklists, respectively. Challenges in meeting these program goals primarily related to the logistics of staff travel to distant health center sites for supervisory purposes and staff shortages for vacations, illness and other limitations in human capital. Nevertheless, our implementation program still facilitated significant improvement in service user symptoms and functional disability. Previous literature has suggested that the development and application of deliberate strategies resulting in evidence-based practice implementation success may be effective across many different clinical innovations and guidelines. [30] Our relatively high degree of implementation success may be related to the previously established feasibility and acceptability of our bundle of implementation strategies within the Rwandan primary care context. [16–17] As task sharing in mental health care also requires complex changes in clinical practice and in the organization of care delivery beyond health provider interventions, the success of MESH MH may also be due to its multifaceted implementation focus on both provider and system level change. [31] Further studies are needed to examine MESH MH with reference to theories of change, as well as potentially to quantify the impact of the individual strategies within the MESH MH program on care delivery outcomes.

Our clinical checklist scores used for audit and feedback to primary care nurses raise another salient question for task sharing in mental health care: As the tasks of evidence-based care packages are distributed among non-specialist care providers, how can basic standards of care for these tasks be encouraged, taught, mentored and feasibly delivered in complex, resource-limited health settings? Our clinical checklist items covered a basic set of assessment and treatment objectives and were developed in context as skills and tasks were reassigned to primary care nurses at health centers. In our study, after nine months of supervision, primary nurses significantly increased their overall performance improvement from 47 to 80% of checklist items scored correctly. For checklist items that focused on treatment initiation at intake, however, including both pharmacologic and non-pharmacologic treatment, improvements occurred but did not reach significance. This could be because our sample size was smaller for these items. However, these smaller improvements and variations in scores over time particularly in psychoeducation and basic non-pharmacologic interventions skills such as behavioral activation, could have been due to multiple factors, including provider knowledge,

understanding of symptom targets for non-pharmacologic treatment components, or logistical concerns such as time available per service user. It is also possible that the training and supervision provided did not engage providers deeply enough on the specifics of good psychosocial and psychotherapeutic (non-pharmacologic) care. The observed variations in checklist item completion reinforce the need to focus on continuous quality improvement of primary care nurse clinical skills to foster sustained basic care package delivery, particularly basic psychosocial and psychotherapeutic care quality. The quality improvement of such care should also reinforce recovery-oriented models for people living with severe mental disorders, and incorporate more and improved person-centered components including self-care interventions and interventions designed to optimize decision-making capacity for service users. The establishment of basic care standards must also recognize that provider burnout can be significant in systems wherein lesser trained individuals are given additional tasks, especially as care is scaled and providers may be taking on more and unfamiliar tasks. [32] The iterative development of achievable standards and competencies as tasks are shared with non-specialist providers, adapted over time as the task shared mental health system matures, may be the most appropriate solution. This is especially important as the mental health services available at health centers are expanded to include the delivery of formal psychotherapies designed for delivery by non-specialist providers.

This study makes several contributions. First, we provide a prospective description of significant improvements in a variety of service user outcomes from the scaled delivery of task shared mental health care packages in a resource-limited, government-run primary care setting. While the findings are limited to four health centers among nineteen in one rural district, they are highly encouraging with regard to the potential for national scale in Rwanda specifically as the government-supported infrastructural elements in the other 41 districts are similar, as well as being promising for application of the model to other settings. These data are important as very limited clinical effectiveness data for mental health task-sharing in such settings exist. Research evidence for task sharing in mental health generally focuses on the efficacy of specific pharmacologic or psychological interventions for certain disorders and rarely evaluates the impact of implementation models designed for scaling task-shared mental health care within real-world settings. Such research trials may have limited external validity as well, as there is often an infusion of financial and human resources into a research endeavour which are not sustained outside of the trial or if so, are not integrated within functional primary care settings. Our data instead report on outcomes for service users receiving care from decentralized mental health services in a real-world government setting of care, supported by a locally designed, innovative, evidence-based implementation model. This may increase the salience of our outcomes to policy makers and health care system planners, in addition to health care providers, who are responsible for improving mental health care outcomes in their own resource-limited settings.

Our study also specifically articulates each component of an implementation strategy outlined by our multifaceted implementation model, aimed at both provider and system levels. These types of strategies are critical for translating evidence-based interventions for task sharing in mental health care into reality; yet such strategies are rarely studied or reported on in low-resource settings. [33] Our strategies are consistent with the growing evidence base in the literature for effective implementation of evidence-based practices, [34–35] and our successful implementation model could facilitate stakeholders to develop similar multifaceted, multilevel implementation models that are tailored to local context, in order to bridge the gap between evidence and practice for mental health care packages in resource limited settings. Further research is needed to rigorously assess the efficacy of the MESH MH model for scaling task-shared mental health services in multiple settings, as well as to assess its sustainability over time.

We evaluated whether the MESH MH program contributed to improved clinical and functional outcomes among service users participating in the program, but we did not include a control group. This is consistent with the implementation science approach of the study, and other potential comparisons—such as enrollment of a contemporary comparison population receiving 'care as usual', or randomized assignment to delayed care—would have been ethically problematic in a location where there are few alternative opportunities for access to quality mental health care. It is possible that the clinical improvements observed in the study may be due to underlying secular trends or the natural course of illness in our population rather than the initiation of program implementation; however, these explanations are less likely given that clinical improvements occurred relatively quickly for most service users and were sustained throughout the study period. Our robust implementation fidelity findings strengthen the plausibility that observed clinical changes can be attributed to services supported by the MESH MH program.

Another limitation is that MESH MH has been used and tested for mental health care scale up to public health centers in only one district in Rwanda—one of 3 districts (out of 42 nationally) that are considered very well-supported by PIH. The availability of logistical and financial support for utilizing the MESH MH implementation strategies may have resulted in a higher impact of the program, including higher fidelity to implementation goals as well as service user outcomes. To address this limitation, current new efforts involve incorporating the MESH MH model within districts not closely supported by PIH, led directly by the Rwanda MoH, with plans for eventual national scale-up to all health centers if effective. MESH MH was successfully used to scale up primary care delivered mental health packages for severe mental disorders in one rural district, yet whether implementation results can be nationally scaled remains to be determined. A costing analysis of MESH MH would have important utility to help to determine whether the operational costs of MESH MH can be absorbed within the scope of the MoH budget dedicated to local health facilities.

## Conclusion

The provision of safe, effective, evidence-based and culturally relevant mental health services at the health-center (primary care) level, linked effectively to functional community health worker networks to provide support at a household level, and to district hospitals that can provide quality higher level care for more severe presentations of illness, represents the holy grail of global mental health. We adapted a multifaceted implementation program initially designed for the decentralization of HIV/AIDS care to facilitate the scale up of health center-delivered packages of care for severe mental disorders in one rural district in Rwanda. Our program implementation was associated with improvements in clinical assessment and treatment delivered by primary care nurses, as well as significant improvements in clinical outcomes and household economic status of service users receiving care at select health centers supported by the program. The results of our work demonstrate that MESH MH represents a promising platform to reduce the evidence to practice gap for mental health care package delivery by non-specialist providers in resource limited settings. Such strategies are imperative in order to reduce the burden of mental disorders across the globe. Further research is needed to determine whether MESH MH could be applied in multiple settings in order to bring evidence-based care packages for delivery by non-specialist providers, to scale.

## Supporting information

**S1 File. MESH MH evaluation proposal, Rwanda National Ethics Committee, July 2014.** (PDF)

**S2 File. Evaluating process and clinical outcomes of a primary care mental health integration project in rural Rwanda: A prospective mixed-methods protocol.**
(PDF)

**S3 File. Trend statement checklist, MESH MH evaluation.**
(PDF)

## Acknowledgments

We would like to acknowledge Roger Levy, PhD for his input on data visualization.

## Author Contributions

**Conceptualization:** Stephanie L. Smith, Molly F. Franke, C. Nancy Misago, Anatole Manzi, Giuseppe J. Raviola.

**Data curation:** Stephanie L. Smith, Beatha Nyirandagijimana, Robert Bienvenu, Eugenie Uwimana, Clemence Uwamaliya, Jean Sauveur Ndikubwimana, Robyn A. Osrow, Rajen Aldis, Shinichi Daimyo.

**Formal analysis:** Stephanie L. Smith, Molly F. Franke, Beatha Nyirandagijimana, Sidney Atwood.

**Funding acquisition:** Stephanie L. Smith, Tharcisse Mpunga, Shinichi Daimyo, Yvonne Kayiteshonga, Giuseppe J. Raviola.

**Investigation:** Stephanie L. Smith, Christian Rusangwa, Hildegarde Mukasakindi, Beatha Nyirandagijimana, Robert Bienvenu, Eugenie Uwimana, Clemence Uwamaliya, Jean Sauveur Ndikubwimana, Sifa Dorcas, C. Nancy Misago, Jean Damascene Iyamuremye, Jeanne d'Arc Dusabeyezu, Achour A. Mohand, Robyn A. Osrow, Alexandra Rose, Sarah Coleman, Anatole Manzi, Yvonne Kayiteshonga, Giuseppe J. Raviola.

**Methodology:** Stephanie L. Smith, Molly F. Franke.

**Project administration:** Stephanie L. Smith, Christian Rusangwa, Hildegarde Mukasakindi, Tharcisse Mpunga, C. Nancy Misago, Jean Damascene Iyamuremye, Jeanne d'Arc Dusabeyezu, Robyn A. Osrow, Rajen Aldis, Shinichi Daimyo, Alexandra Rose, Anatole Manzi, Yvonne Kayiteshonga, Giuseppe J. Raviola.

**Resources:** Stephanie L. Smith, Tharcisse Mpunga, C. Nancy Misago, Jean Damascene Iyamuremye, Jeanne d'Arc Dusabeyezu, Achour A. Mohand, Alexandra Rose, Sarah Coleman, Yvonne Kayiteshonga, Giuseppe J. Raviola.

**Software:** Molly F. Franke, Sidney Atwood.

**Supervision:** Stephanie L. Smith, Christian Rusangwa, Hildegarde Mukasakindi, Beatha Nyirandagijimana, Robert Bienvenu, Eugenie Uwimana, Clemence Uwamaliya, Jean Sauveur Ndikubwimana, Sifa Dorcas, Tharcisse Mpunga, Robyn A. Osrow, Rajen Aldis, Anatole Manzi, Yvonne Kayiteshonga, Giuseppe J. Raviola.

**Validation:** Stephanie L. Smith, Molly F. Franke, Christian Rusangwa, Hildegarde Mukasakindi, Beatha Nyirandagijimana, Robert Bienvenu, Sidney Atwood, Rajen Aldis, Alexandra Rose, Sarah Coleman.

**Visualization:** Stephanie L. Smith, Molly F. Franke, Sidney Atwood.

**Writing – original draft:** Stephanie L. Smith, Molly F. Franke.

**Writing – review & editing:** Stephanie L. Smith, Molly F. Franke, Christian Rusangwa, Hildegarde Mukasakindi, Beatha Nyirandagijimana, Robert Bienvenu, Eugenie Uwimana, Clemence Uwamaliya, Jean Sauveur Ndikubwimana, Sifa Dorcas, Tharcisse Mpunga, C. Nancy Misago, Jean Damascene Iyamuremye, Jeanne d'Arc Dusabeyezu, Achour A. Mohand, Sidney Atwood, Robyn A. Osrow, Rajen Aldis, Shinichi Daimyo, Alexandra Rose, Sarah Coleman, Anatole Manzi, Yvonne Kayiteshonga, Giuseppe J. Raviola.

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
