## [Decision Letter · Decision Letter 0]

4 Nov 2019

PONE-D-19-23635

Outcomes of a primary care mental health implementation program in rural Rwanda: a quasi-experimental implementation-effectiveness study

PLOS ONE

Dear Dr. Smith,

Thank you for submitting your manuscript to PLOS ONE. After careful consideration, we feel that it has merit but does not fully meet PLOS ONE’s publication criteria as it currently stands. Therefore, we invite you to submit a revised version of the manuscript that addresses the points raised during the review process.

We would appreciate receiving your revised manuscript by Dec 19 2019 11:59PM. To enhance the reproducibility of your results, we recommend that if applicable you deposit your laboratory protocols in protocols.io, where a protocol can be assigned its own identifier (DOI) such that it can be cited independently in the future. For instructions see: http://journals.plos.org/plosone/s/submission-guidelines#loc-laboratory-protocols

We look forward to receiving your revised manuscript.

Kind regards,

Thach Duc Tran, M.Sc., Ph.D.

Academic Editor

PLOS ONE

Journal Requirements:

1.When submitting your revision, we need you to address these additional requirements.

"This study was generously funded by the Abundance Foundation and Rick and Nancy Moskovitz. The funders had no role in study design, data collection and analysis, decision to publish, or preparation of the manuscript. "

We note that one or more of the authors are employed by a commercial company: 'Partners In Health'.

Additional Editor Comments (if provided):

Please provide a statement on how sample size was determined. Explain why no control group was involved and the possible effects to the main findings.

Reviewers' comments:

Reviewer's Responses to Questions

**Comments to the Author**

1. Is the manuscript technically sound, and do the data support the conclusions?

Reviewer #1: Partly

Reviewer #2: Yes

2. Has the statistical analysis been performed appropriately and rigorously? 

Reviewer #1: Yes

Reviewer #2: Yes

3. Have the authors made all data underlying the findings in their manuscript fully available?

Reviewer #1: Yes

Reviewer #2: Yes

4. Is the manuscript presented in an intelligible fashion and written in standard English?

Reviewer #1: Yes

Reviewer #2: Yes

5. Review Comments to the Author

Reviewer #1: This paper reports a well conducted implementation study of a novel primary mental health care intervention, undertaken in challenging circumstances in rural Rwanda. The authors explain how the intervention was developed pragmatically on the back of existing prmary health care programmes, and clearly specify key elements of the delivery package (training, mentoring, performance audit and QI project collaboration). They provide a convincing rationale for undertaking a before- and after-study without controls or comparison groups. Their primary (GHQ-12 and WHO-DAS) and secondary outcomes (including income generation) are reasonable. The findings are, in general, clearly described and suitable caveats are in place regarding their clinical significance and their generalisability.

I have three comments/queries:

1. The abstract needs to be rewritten to distinguish clearly between methods and results. The methods currently describes a cohort of 146 adults, but then the results begin by referring to 2239 service users. My understanding of the paper is that the 146 were a sub-set of the 2239. This needs to be clarified. I would expect the methods section of the abstract to summarise the methods section of the main paper, including key elements of the delivery programme, which it currently does not do. (The main text is clear on this distinction). The concluding section of the abstract should briefly note methodological limitations of a pre-post design.

2. The discussion of limitations in the main paper could expand on the interesting findings of differences in records regarding assessment vs interventions, and especally why psychoeducation interventions did not improve and indeed appear to drop off. I wonder if these types of intervention are more intensive/challenging for the nurses to deliver, and hence would benefit from greater input at training/mentoring phases.

3. 83% is impressive for 6 month articipant follow up, in any context. What proportion of particiapnts needed support/encouragement from CHWs i.e. how significant was the role of CHWs in this whole delivery process?

Reviewer #2: I really enjoyed reading this paper. It is a well written, structured paper that will appeal to the journals authorship. Topically it is important and work on global health solutions has an increasing national and international focus.

The paper is of particular importance as it focusses on a multifaceted implementation programme which was initially designed to integrate HIV/AIDS care into primary care and have adapted this methodology for severe mental disorders and epilepsy in rural Rwanda. Such knowledge and methods are of crucial importance if we are ever to bridge the know-do gap and commend the authors on their work. This paper makes a significant contribution to the literature.

The authors have acknowledged the methodological implications of the use of pre-post design.

A few issues that I think require addressing are;

• Justifcation/rationale for the inclusion of schizophrenia; bipolar disorder; major depressive disorder and epilepsy is needed. Although the authors have justified the inclusion of mental health and epilepsy – justification of how the 3 specific mental health disorders were selected is needed (eg burden, disability, prevalence, policy drivers or multiple reasons).

• There appears to be absence of involvement of the patients and families in the intervention both at devising the multi-faceted implementation programme and in the delivery. For example in the ‘Treatment planning: non-medication based and the Treatment planning: medication management sections in Table 3. Subset of Checklist Indicators there is a noticeable lack of patient and family involvement – so it is framed top down ie Tell the patient how the medication might help, tell the patient - Tell the patient about potential side effects. The absence of patient involvement, self-care- management and particularly shared decision making is noticeable throughout the paper .

• I may have missed it but I was unable to identify how many participants were triaged and e and referred to specialist mental health care for acute or complex needs.

6. PLOS authors have the option to publish the peer review history of their article (what does this mean?). If published, this will include your full peer review and any attached files.

Reviewer #1: Yes: Christopher Dowrick

Reviewer #2: Yes: Karina Lovell

---

## [Author Response · Author response to Decision Letter 0]

17 Jan 2020

Responses to the Journal’s requirements (also addressed in the Cover Letter and Responses to Reviewers): 

1) As a point of clarification, the organization Partners In Health is a non-profit 501(c)(3) organization, not a commercial entity as the editor mentions in point 1 of the ‘Journal Requirements’. We have edited our Funding Statement and Competing Interests statement as requested by the Editor, but want to make sure that our organization is not misrepresented as a commercial company. 

Responses to the Editor’s comments:

1) We have added a section on how our sample size was determined (entitled Sample Size in the Methods section). We have also added into the Abstract a statement on the primary limitation of the study (lack of a control group), and have discussed this limitation and possible effects of the limitation in paragraph 8 of the Discussion section, particularly that the clinical improvements observed in the study may be due to underlying secular trends or the natural course of illness in our population rather than the initiation of program implementation. 

Responses to Reviewer #1:

1) As requested, we have rewritten the abstract to distinguish clearly between methods and results. We have included a brief description of the key elements of the MESH MH implementation program within the methods section, as noted by the reviewer. In addition, we clarify that the population participating in the outcome evaluation is a group of consecutive adults presenting to the health centers over a nine month period—a subset of the total number of patients seen over the two year scale-up period. We have stated the primary limitation of the study in the conclusion of the abstract. Please also refer to the tracked changes version of the revised manuscript. 

2) In comment #2, the reviewer asks for expansion of the discussion on the differences among checklist item completion subgroups (assessment versus intervention). These differences, and specifically the variable scores of the nurses on non-pharmacologic interventions, as mentioned by the reviewer, is discussed relatively extensively in paragraph 5 of the Discussion section. We have now made some minor edits to the section to make more clear that we are discussing the issues put forth by the reviewer. If additional expansion is still requested beyond our edits, please give further detail on what elements we should expand on given that this section of the Discussion is relatively lengthy already.

3) We have added a comment in the Results section highlighting the role of CHWs for those who continued in care after six months. 

Responses to Reviewer #2:

1) In the Methods, in the section ‘Evidence-based Care Packages’, we have made clearer how the initial areas of clinical focus (psychosis, severe depression and epilepsy) were chosen. The specific clinical areas were determined both by perceived burden of illness attributable to these disorders within the district, as well by the stated clinical priorities of health system leaders in Burera district during the early stages of MESH MH development. 

2) We agree with the reviewer that in many ways, the program was designed from a health system perspective in mind—namely that the program components were focused on improving quality of care delivered by non-specialist providers, and more could be added to the program around supported decision-making with service users. We have added some text in paragraph 8 of the Discussion section about next steps and need for program improvements vis a vis effective participation in program development by service users. 

3) In the Results, in the section entitled ‘Evaluation of participants and follow-up’ we have clarified how many individuals (four) were excluded from the program evaluation due to acute psychiatric or medical needs.

---

## [Decision Letter · Decision Letter 1]

27 Jan 2020

Outcomes of a primary care mental health implementation program in rural Rwanda: a quasi-experimental implementation-effectiveness study

PONE-D-19-23635R1

Dear Dr. Smith,

We are pleased to inform you that your manuscript has been judged scientifically suitable for publication and will be formally accepted for publication once it complies with all outstanding technical requirements.

With kind regards,

Thach Duc Tran, M.Sc., Ph.D.

Academic Editor

PLOS ONE

Additional Editor Comments (optional):

Reviewers' comments:

Reviewer's Responses to Questions

**Comments to the Author**

1. If the authors have adequately addressed your comments raised in a previous round of review and you feel that this manuscript is now acceptable for publication, you may indicate that here to bypass the “Comments to the Author” section, enter your conflict of interest statement in the “Confidential to Editor” section, and submit your "Accept" recommendation.

Reviewer #1: All comments have been addressed

Reviewer #2: All comments have been addressed

2. Is the manuscript technically sound, and do the data support the conclusions?

Reviewer #1: Yes

Reviewer #2: Yes

3. Has the statistical analysis been performed appropriately and rigorously? 

Reviewer #1: Yes

Reviewer #2: Yes

4. Have the authors made all data underlying the findings in their manuscript fully available?

Reviewer #1: Yes

Reviewer #2: Yes

5. Is the manuscript presented in an intelligible fashion and written in standard English?

Reviewer #1: Yes

Reviewer #2: Yes

6. Review Comments to the Author

Reviewer #1: (No Response)

Reviewer #2: (No Response)

7. PLOS authors have the option to publish the peer review history of their article (what does this mean?). If published, this will include your full peer review and any attached files.

Reviewer #1: No

Reviewer #2: Yes: Karina Lovell

---

## [Editor Report · Acceptance letter]

11 Feb 2020

PONE-D-19-23635R1 

Outcomes of a primary care mental health implementation program in rural Rwanda: a quasi-experimental implementation-effectiveness study 

Dear Dr. Smith:

I am pleased to inform you that your manuscript has been deemed suitable for publication in PLOS ONE. Congratulations! Your manuscript is now with our production department. 

With kind regards,

on behalf of

Dr. Thach Duc Tran 

Academic Editor

PLOS ONE